# Are Matrix Metalloproteinase-9 and Tissue Inhibitor of Metalloproteinase-1 Useful as Markers in Diagnostic Management of Children with Newly Diagnosed Ulcerative Colitis?

**DOI:** 10.3390/jcm11092655

**Published:** 2022-05-09

**Authors:** Aleksandra Czajkowska, Katarzyna Guzinska-Ustymowicz, Anna Pryczynicz, Dariusz Lebensztejn, Urszula Daniluk

**Affiliations:** 1Department of Pediatrics, Gastroenterology, Hepatology, Nutrition and Allergology, Medical University of Bialystok, 17 Waszyngtona Street, 15-274 Bialystok, Poland; dariusz.lebensztejn@umb.edu.pl (D.L.); urszula.daniluk@umb.edu.pl (U.D.); 2Department of General Pathomorphology, Medical University of Bialystok, 15-089 Bialystok, Poland; katarzyna.guzinska-ustymowicz@umb.edu.pl (K.G.-U.); anna.pryczynicz@umb.edu.pl (A.P.)

**Keywords:** ulcerative colitis, metalloproteinase, Tissue Inhibitor of Metaloproteinase, children

## Abstract

Matrix Metaloproteinase-9 (MMP-9) and Tissue Inhibitor of Metaloproteinase-1 (TIMP-1), enzymes involved in tissue remodelling, have been previously reported to be overexpressed in the colonic mucosa of patients with Ulcerative colitis (UC). The aim of this study was to determine the relation of MMP-9 and TIMP-1 with UC phenotypes, the disease activity index and routinely tested inflammatory markers in newly diagnosed paediatric patients. The study group comprised 35 children diagnosed with UC and 20 control groups. Serum and faecal concentrations of MMP-9 and TIMP-1 were estimated using enzyme-like immunosorbent assay kits and correlated to the disease activity index (Paediatric Ulcerative Colitis Activity Index, PUCAI), UC phenotype (Paris Classification), inflammatory markers and endoscopic score (Mayo score). Children with UC presented with significantly higher serum and faecal concentrations of studied markers compared to the control group. Both serums, MMP-9 and TIMP-1, were higher in children with more extended and severe lesions in the colon. Furthermore, serum MMP-9 correlated with the Mayo score, Paris classification and C-reactive protein (CRP) levels. Serum TIMP-1 showed correlation with PUCAI, Paris Classification, CRP levels and the erythrocyte sedimentation rate. Serum and faecal levels of MMP-9 and TIMP-1 are useful in discriminating UC patients and non-invasive assessments of disease phenotypes. It seemed that simultaneous measurement of these proteins in combination with other common markers of inflammation could be applied in clinical practice.

## 1. Introduction

Ulcerative colitis (UC) is a chronic inflammatory disease characterised by episodes of recurrence and remission. Due to non-specific symptoms and the lack of highly specific biomarkers, UC often evades diagnosis for a long time [1]. Early diagnosis is crucial in paediatric patients to avoid long-term complications, such as delayed puberty, malnutrition and growth failure, insufficient response to treatment and an increased need for surgery.

The diagnosis depends upon series of tests, including invasive procedures such as colonoscopies. In recent years, there has been an on-going search for non-invasive markers of intestinal inflammation, which could reduce the use of endoscopy in diagnosis and evaluation of the disease course. The limitation of endoscopic examination is particularly important in children, due to the high invasiveness of the procedure and the need to perform endoscopies under general anaesthesia. Nowadays, among the most commonly used markers is faecal calprotectin (FCal); however, it is not specific for UC, as its elevated concentration is observed in almost all inflammatory diseases of the lower gastrointestinal (GI) tract [2,3,4].

Matrix Metaloproteinase-9 (MMP-9) and Tissue Inhibitor of Metaloproteinase-1 (TIMP-1) are enzymes involved in tissue remodeling and their overexpression in colonic mucosa of patients suffering from UC have been previously reported [5,6,7,8]. Additionally, increased levels of both MMP-9 and TIMP-1 have been demonstrated in the urine, blood and stool of UC patients [9,10,11,12]. Although those enzymes are well studied in adults, there is no sufficient research in paediatric patients suffering from UC; thus, we decided to study indicated markers in this group.

The aim of this study was to determine the relation of MMP-9 and TIMP-1 with UC phenotypes, a disease activity index and routinely tested inflammatory markers in newly diagnosed paediatric patients. The secondary aim was to evaluate their clinical value as non-invasive biomarkers.

## 2. Materials and Methods

### 2.1. Study Group

The study group comprised 35 patients with newly diagnosed UC between the years 2014 and 2019. The diagnosis was based on the European Society for Paediatric Gastroenterology, Hepatology and Nutrition (ESPGHAN) guidelines, including endoscopic and histological criteria [13]. The disease activity was estimated according to the Paediatric Ulcerative Colitis Index (PUCAI) and rated as: mild or inactive (<35 points), moderate (35–64 points) or severe (>64 points). Endoscopic disease activity was evaluated according to Mayo Endoscopic score as: 0—normal colonic mucosa, 1—erythema, decreased vascular pattern, mild friability, 2—marked erythema, erosions, no vascular pattern, friability, 3—ulceration, spontaneous bleeding. The extent of the lesions in the colon was assessed according to the Paris classification as follows: E1—proctitis, E2—distal to splenic flexure colitis, E3—extensive, distal to hepatic flexure and E4—pancolitis.

The control group (Ctr) consisted of 20 children with excluded gastrointestinal inflammatory disease and normal FCal level (<50 ug/g).

Patients were not being treated at the time of diagnosis and collection of samples for tests.

The study protocol was approved by the local Ethics Committee (R-I-002/308/2014, R-I-002/294/2018) and conformed to the tenets of the Declaration of Helsinki. Written informed consent was obtained from the parents of all the study participants.

### 2.2. Samples Collection and Measurements of MMP-9, TIMP-1, Routine Blood Tests and FCal

All patients were fasting on the day of sample collection. Blood samples were drawn by venous puncture. Sera were obtained from clotted and centrifuged blood. Faecal samples were collected during hospitalization. Serum and faecal samples were stored at −80 C until analysis.

Data on C-reactive protein (CRP), complete blood count, erythrocyte sedimentation rate (ESR), albumin and FCal was obtained from the patient’s medical records.

Both MMP-9 and TIMP-1 serum and faecal concentrations were estimated using enzyme –like immunosorbent assay (ELISA) kits (Quantikine; R&D system, Abingdon, UK) according to the manufacturer’s protocol.

### 2.3. Expression of MMP-9 and TIMP-1 in Colonic Tissue

The intestinal biopsies were taken during the colonoscopy (*n* = 19). After fixing with 10% phosphate-buffered saline buffered formalin, and embedding in paraffin, tissue samples were cut sagitally into 4 μm sections and incubated with anti-human antibodies: mouse monoclonal Matrix Metalloproteinase 9 antibody (clone 15 W2; Novocastra, Newcastle, UK; dilution 1:80) or mouse monoclonal Tissue Inhibitor of Matrix Metalloproteinase 1 antibody (clone 6F6a; Novo- castra, Newcastle, UK; dilution 1:150) for 1 h at room temperature. Following the reaction in the streptavidin–biotin system (Biotinylated Secondary Antibody, Streptavidin HRP; Novocastra, Newcastle, UK), the antigen–antibody complex was visualised with the use of chromogen 3,3—diaminobenzidine (DAB; Novocastra, Newcastle, UK). All stained specimens were assessed by light microscopy (Olympus BX45, Olympus Corporation, Tokyo, Japan) by an independent pathologist who was blind to all clinical information. The staining intensity was assessed as negative (0), weak (1+), moderate (2+) or strong (3+), and the extent of the staining was classified as the percentage of positive staining areas scored on a scale of 0 to 4 as follows: 0, 0%; 1, 1–10%; 2, 11–30%; 3, 31–50%; 4, ≥51%. The final staining score was the sum of the staining- intensity and staining-extent scores.

### 2.4. Statistical Analysis

Data analysis was performed using Statistica 13 software. The significance of the difference in data was evaluated with the non-parametric Mann–Whitney U-test and the Kruskal–Wallis test by ranks supplemented with the post-hoc test. Spearman’s correlation test was used to analyse the correlations between variables. The diagnostic value of MMP-9 and TIMP-1 concentration was estimated using receiver operating characteristic (ROC) curve analyses. To calculate the sensitivity, the specificity and the accuracy of MMP-9 and TIMP-1, the cut-off values were selected based on the Youden index maximization criterion. *p* < 0.05 was considered as statistically significant.

## 3. Results

### 3.1. Patient’s Background

The study involved 35 children with UC and 20 children as a Ctr. The demographic and clinical features of included patients are presented in Table 1. There were no significant differences in sex and age between the groups. Disease activity according to the PUCAI was scored as follows: severe in 6 (17.2%) patients, moderate in 15 (42.8%) patients and mild in 14 (40%) patients.

Endoscopic lesions were assessed according to Mayo Endoscopic score as severe in 16 patients (45.7%), moderate in 15 (42.8%) patients and mild in 4 (11.4%) patients. The extent of the disease was assessed as E1 in 7 (20%) patients, E2 in 12 (34.3%) patients, E3 in 10 (28.6%) patients and E4 in 6 (17.1%) patients.

### 3.2. Diagnostic Utility of MMP-9 and TIMP-1 in Distinguishing between UC and Ctr

The levels of MMP-9 and TIMP-1 as well as selected inflammatory markers in UC and Ctr are presented in Table 2. Patients with UC presented with significantly higher MMP-9 serum and faecal levels compared to controls (Table 2, Figure 1a,b). Also, concentration of serum and faecal TIMP-1 in patients with UC were significantly elevated when compared to controls (Table 2, Figure 1c,d).

The ROC analysis was performed in order to assess the diagnostic value of studied markers. The best results were obtained for faecal MMP-9, which with the cut-off level of 33.5 ng/mL discriminated patients with UC from Ctr with 94.3% sensitivity and 72.2% specificity (AUC = 0.925) (Table 3). For faecal TIMP-1, the AUC was 0.881 with 74.3% sensitivity and 95% specificity at the cut-off level 5.8 ng/mL. To distinguish patients with UC from controls, the cut-off level of serum TIMP-1 was determined at 154.2 ng/mL, with the AUC 0.809, 74.3% sensitivity and 80 % specificity.

Of the selected markers, only serum MMP-9 was proven to have the lowest diagnostic value with an AUC of 0.712 at the cut-off value of 304.1 ng/mL.

### 3.3. Analysis of the Relationship between MMP-9 Concentration in Serum and Feces on Disease Activity and UC Phenotype

Next, we analysed whether there were differences in MMP-9 and TIMP-1 concentrations between groups of UC patients distinguished on the basis of endoscopic assessment (Paris classification, Mayo Endoscopic Score) and a disease activity index (PUCAI).

No statistical differences in MMP-9 concentrations were observed in patients with different disease activity. However, taking into consideration severity of endoscopic lesions, there were significant differences in faecal MMP-9 among Mayo score groups (Figure 2a, *p* < 0.037 Kruskall–Wallis test). Surprisingly, faecal MMP-9 concentration was significantly higher in patients with Mayo 1 when compared to patients with Mayo 2 using the post-hoc test (Figure 2a, *p* = 0.031). In contrast, serum MMP-9 was observed to be higher in patients with more severe lesions (Mayo 3) (Figure 2b, *p* = 0.04). Interestingly, no differences were discovered when considering the extent of the lesions in Paris Classification, but patients with right-sided (proximal colitis, E3 or E4) lesions had statistically higher serum MMP-9 concentration than patients with left-sided (distal colitis, E1 or E2) lesions (median 783.6 ng/mL vs. 492.3 ng/mL, *p* = 0.01, Figure 3a). It should be mentioned that patients with more severe endoscopic lesions more often had the right colon affected also. Additionally, we found significant differences in levels of the other inflammatory markers between these two groups (Table 4). Therefore, we decided to check whether serum MMP-9 may distinguish children with right-sided (E3, E4) colitis from those with left-sided (E1, E2) colitis. In the ROC analysis, we noted 84.2% specificity and 56.3% sensitivity for 658 ng/mL cut-off level of serum MMP-9 (AUC = 0.75, *p* = 0.003) (Table 5).

### 3.4. Analysis of the Relationship between TIMP-1 Concentration in Serum and Feces on Disease Activity and UC Phenotype

There was no difference in TIMP-1 concentration when comparing children with different activity of the disease (PUCAI).

However, concentration of TIMP-1 differed between the subgroups of patients differing in the severity of endoscopic lesions (Mayo Endoscopic Score), with the highest levels of faecal TIMP-1 in the group of patients with Mayo 1 vs. Mayo 2 (Figure 2c, *p* = 0.003) and serum TIMP-1 in the group of patients with the most severe lesions (Mayo 3) comparing to Mayo 2 (Figure 2d, *p* = 0.01). Taking into consideration the extent (E) of the lesions, serum levels of TIMP-1 were also higher in patients with pancolitis; however, the difference was on the verge of statistical significance (*p* = 0,051, data not shown). Interestingly, when comparing right-sided (E3, E4) colitis with left-sided (E1, E2) colitis, the concentration of TIMP-1 in serum was significantly higher in patients with right-sided lesions (median 215.6 ng/mL vs. 157.6 ng/mL, *p* = 0.006, Figure 3b). We also checked the diagnostic value of serum TIMP-1 in distinguishing children with right-sided (E3, E4) colitis from those with left-sided (E1, E2) colitis. The cut-off level of 210.9 ng/mL discriminated patients with 56.3.3% sensitivity and 89.5% specificity (AUC = 0.77, *p* < 0.000) (Table 5).

No differences in concentration of faecal TIMP-1 were found between the patients distinguished based on disease activity or extent of endoscopic lesions. Moreover, no differences in FCal levels were found in relation to PUCAI, to the severity of endoscopic lesions and to the extent of endoscopic lesions (data not shown).

### 3.5. Correlation between Inflammatory Markers, Disease Activity Index, Endoscopic Score and MMP-9 and TIMP-1

Both serum TIMP-1 and MMP-9 significantly correlated with CRP, PLT and Paris classification (Table 6). Additionally, correlations of serum TIMP-1 with ESR, Hb, albumin and PUCAI were noted. For serum MMP-9, the positive correlation with white blood count (WBC) and Mayo endoscopic score was found.

Interestingly, there was no significant correlations between FCal and both serum and fecal MMP-9 and TIMP-1 levels, respectively.

No correlations were found for both faecal TIMP-1 and MMP-9.

### 3.6. Immunochemistry

There were no statistical differences in final score of intensity and the extent of expression of MMP-9 and TIMP-1 in study group in colon biopsies (Appendix A). Interestingly, the TIMP-1 expression, described as percentage of positive staining areas, was higher than the MMP-9 expression in UC samples (respectively ranging from 30–100% vs. 20–100%, mean value 100% vs. 70%).

There was no correlation between the staining and serum and fecal concentration of studied markers (data not shown).

## 4. Discussion

In recent years metalloproteinases and their inhibitors have been suggested to play a role in pathogenesis and inflammatory processes in IBD [5,6,7,8,9,11,12,14,15,16,17,18,19,20,21,22]. As MMPs and TIMPs are involved in the remodeling of an extracellular matrix, MMP-9 mediates an increase in proinflammatory cytokines and TIMP-1 influences cell functions as a cytokine modulator [23,24]. Moreover, MMP-9 also affects intestinal permeability but does not induce apoptosis of intestinal cells [23,25,26]. In physiological conditions, both the enzymes’ functions are balanced and do not cause tissue damage; however, in the IBD inflammatory environment the overexpression of MMP-9 and TIMP-1 results in their increased activity [27]. The dysregulated MMP-9 function causes diminished cell adhesion, promotes cytokine production and attracts neutrophils to the gut epithelium, thus further attenuating inflammation [25,26,28].

The overexpression of MMP-9 and TIMP-1 in colonic mucosa of UC patients, both in adults and children, was reported in many studies [5,6,7,8]. Although the studies on intestinal tissue in children are promising, there are few reports measurement of MMP-9 and TIMP-1 in various other biological samples (blood, stool, urine) [9,11,12]. In fact, to the best of our knowledge, there is only one study measuring TIMP-1 faecal concentration in the paediatric population [29].

In this study we evaluated MMP-9 and TIMP-1 levels in children with UC and Ctr. As far as we are aware, this is the first simultaneous measurement of TIMP-1 in serum and stool in such group of patients. Our results showing elevated levels of serum TIMP-1 and MMP-9 and faecal MMP-9 were consistent with earlier reports on adult patients and children [8,10,15,16,17,18,19,20,21,22]. Moreover, in children with UC, a decrease in serum TIMP-1 concentration was demonstrated after 4 weeks of glucocorticosteroid (GC) therapy by Makitalo et al. [19]. This finding may suggest the possibility of using serum TIMP-1 measurement in the evaluation of disease activity or treatment efficacy. In our study, serum TIMP-1 correlated well with majority of the inflammatory markers used in clinical practice, comparable to the findings in adults. Wiercinska-Drapalo et al. demonstrated that TIMP-1 levels in blood corresponded well with the degree of endoscopic mucosal injury, disease activity index and clinical activity index, corroborating our results [22]. Our analysis revealed some diagnostic value of both serum TIMP-1 and MMP-9 in distinguishing children with left-sided colitis from those with right-sided colitis. Moreover, a combination of both serums TIMP-1 and MMP-9 with other common inflammatory markers could be useful to predict more extended inflammation in colon. Additionally, we noted the positive correlation of the extent of lesions (according to the Paris classification) with serum TIMP-1 levels, oppositely to the report of Kapsoritakis et al. [18]. Moreover, the authors also found significantly higher serum concentrations of this protein in males than females; nevertheless, this result was also not repeated by our study. Increased levels of TIMP-1 in severe diseases indicate the possibility of using it as a marker of disease exacerbation; however, additional studies on a larger group of patients before and after therapy are required.

Interestingly, even though faecal TIMP-1 did not demonstrate as much of an association with PUCAI and endoscopic evaluation, it turned out to be a better marker in distinguishing UC from Ctr than its serum levels. Soomro et al. also showed increased levels of TIMP-1 in the stool of children with UC and furthermore confirmed its ability to differentiate UC from healthy controls, confirming our results [11,29]. Unfortunately, in our study, faecal TIMP-1 was not associated with either the UC phenotype or disease activity.

Our study found that faecal MMP-9 best distinguished UC from Ctr. The result was comparable to study by Annahazi et al., Farkas et al. in adults and Kolho et al. in children [11,15,16]. Although these studies mention how this marker correlated well with serum CRP, Fcal, disease activity index and endoscopic activity, we have not noted a similar connection. Interestingly, we observed that both faecal MMP-9 and TIMP-1 were significantly higher in patients with mild lesions in the colon, assessed as 1 according to the Mayo endoscopic subscore. Serum MMP-9 concentrations were significantly higher in a group of patients with more intensive lesions, whereas serum TIMP-1 concentrations were not. This result may suggest that in the early stages of mucosal injury, when the lesions are superficial, the permeability of the colon mucosa is still effective and prevents absorption of the released enzyme. Interestingly, Al-Sadi et al. have reported an effect of MMP-9 on increased intestinal permeability, supporting our theory [25,26].

In our study, we have demonstrated that serum MMP-9 correlates with CRP, WBC, PUCAI and Mayo endoscopic scores. Unfortunately, this marker does not discriminate well between UC and Ctr. Our results were confirmed in the study by Kofla-Dłubacz et al., which showed a similar relationship with inflammatory markers and the disease activity index (DAI) [9]. Those results suggest that MMP-9 may not be useful in disease identification but may help with assessment of its activity. However, Makitalo et al., who also reported elevated levels of MMP-9 in blood of children with UC, showed there is no difference in its concentration after therapy with glucocorticosteroid or anti-Tumour Necrosis Factor-α [19]. Another study by Faubion et al. suggested that combination of FCal and serum MMP-9 is more effective in predicting inflammation in UC patients [17]. Since the results are inconclusive, further studies with bigger cohorts of patients are needed to evaluate serum MMP-9 concentrations connection to the activity of the disease. Nonetheless, there is still no specific marker that could be used to monitor disease activity in children. Currently, FCal is the most commonly used marker in the diagnosis and monitoring of IBD [30]. It is derived from the neutrophils infiltrating the intestine and serves as a marker of intestinal inflammation [30]. Although FCal is very sensitive, it is not highly specific to UC as its increased levels are also observed in GI tract infections, polyposis or neoplasms [31,32,33,34]. In our study, faecal concentrations of MMP-9 and TIMP-1 were elevated in UC patients, as was FCal [30]. However, in contrast to FCal concentrations, faecal MMP-9 and TIMP-1 levels varied between patient subgroups differing in the severity of endoscopic lesions. Our results on FCal contrast other reports which show that FCal increases along with the severity of the colon damage [35,36]. On the other hand, in our study, the serum MMP-9 concentrations correlated with the endoscopic score. Additionally, we also noted higher blood inflammatory parameters in the study group than in Ctr, so combining them with a high MMP-9 and TIMP-1 may help to differentiate UC patients not only from Ctr, but also to determine the disease severity. Recently, new promising markers for UC diagnosis in children have been reported, such as B cell-activation factor (BAFF) or lipid C16:0—Lactosylceramide [37,38]. Therefore, maybe the combination of several markers could be more useful in monitoring of the disease and predicting the outcome than the single parameter.

The study has some limitations, the most important of which is the small number of patients enrolled in it, which could have affected obtained results, especially the correlation between the studied proteins and other markers. The calculation of sample size was performed; however, it was a single-center study and within 5 years the number of newly diagnosed patients was lower than the calculated sample size. Our study was an exploratory study, but not a confirmatory study where achievable effect sizes and variance can be estimated from previous exploratory evidence. We are aware that our results may be subject to type II error (low test sensitivity). That is why we did not interpret non-significant statistical results as an underlying true lack of differences. We are aware that our results need to be confirmed in further studies preceded by the calculation of sample size and test power. There is a lack of the assessment of studied markers in children with other intestinal diseases, which would significantly improve the relevance of the results.

The strength of this study is the well-composed study group, consisting of newly diagnosed patients with no differences in age and sex to the Ctr group. Moreover, blood and stool samples for assessing TIMP-1 and MMP-9 levels were obtained prior to induction therapy, which could have an impact on the results. Finally, this was the only study to assess TIMP-1 levels in both serum and faeces of children with IBD, and one of two studies to assess faecal concentration of TIMP-1 in the pediatric population.

## 5. Conclusions

In conclusion, our study showed the diagnostic potential of both TIMP-1 and MMP-9 measurements, suggesting their clinical utility in children with UC. It seemed that simultaneous measurement of these proteins both in serum and stool in combination with other common markers of inflammation could be applied in clinical practice; however, it is necessary to evaluate their concentrations in remission of UC induced by therapy and in other inflammatory GI pathologies in children.

## Figures and Tables

**Figure 1 jcm-11-02655-f001:**
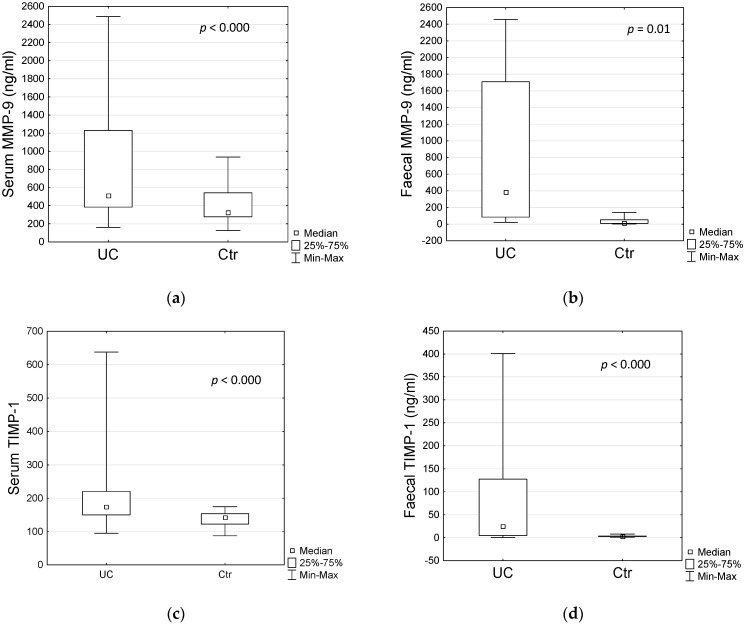
Serum and faecal MMP-9 (**a**,**b**) and TIMP-1 (**c**,**d**) concentrations in UC and Ctr groups. Statistical analysis by Mann–Whitney U test. UC—ulcerative colitis, Ctr—control group, MMP-9—matrix metalloproteinase-9, TIMP-1—tissue inhibitor of metalloproteinase-1.

**Figure 2 jcm-11-02655-f002:**
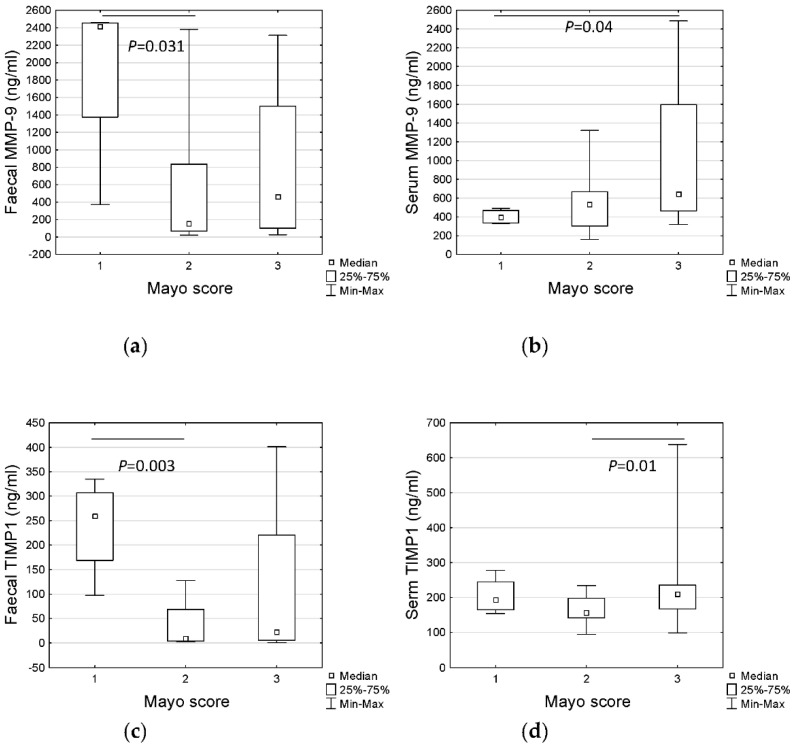
Differences in MMP-9 (faecal—(**a**), serum—(**b**)) and TIMP-1 (faecal—(**c**), serum—(**d**)) concentration based on the severity of the lesions in the colonic mucosa (Mayo Endoscopic Score). MMP-9—matrix metalloproteinase-9, TIMP-1—tissue inhibitor of metalloproteinase-1, Mayo—Mayo Endoscopic score: 1—erythema, decreased vascular pattern, mild friability, 2—marked erythema, erosions, no vascular pattern, friability, 3—ulceration, spontaneous bleeding. Statistical analysis performed using Kruskall–Wallis test supplemented with the post-hoc test.

**Figure 3 jcm-11-02655-f003:**
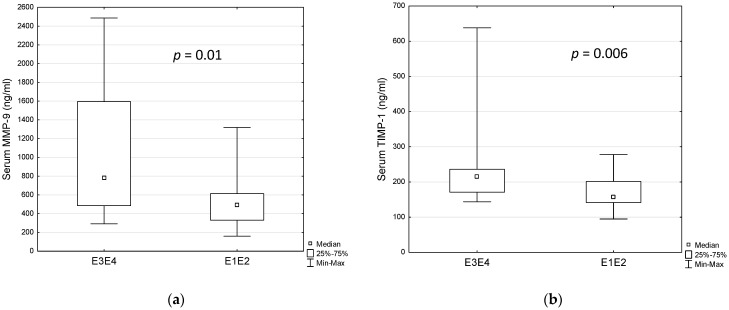
Differences in serum MMP-9 (**a**) and TIMP-1 (**b**) concentration regarding the extent of the disease (Paris classification). Statistical analysis by Mann–Whitney U test. MMP-9—matrix metalloproteinase-9, TIMP-1—tissue inhibitor of metalloproteinase-1.

**Table 1 jcm-11-02655-t001:** Clinical characteristics of children with UC and Ctr.

	UC	Ctr	*p*
Girls n (%)	17 (48.5%)	9 (45%)	NS
Boys n (%)	18 (51.5%)	11 (55%)	NS
Age median	14 (4–17)	14.5 (4–17)	NS
PUCAI median Mild n (%) Moderate n (%) Severe n (%)	42.5 (0–80) 14 (30%) 15 (42.8%) 6 (17.2%)	N/A	-
Mayo endoscopic score, n (%)			
0	0	N/A	-
1	4 (11.4%)	N/A	-
2	15 (42.8%)	N/A	-
3	16 (45.7%)	N/A	-
Paris classification, n (%)			
E1	7 (20%)	N/A	-
E2	12 (34.3%)	N/A	-
E3	10 (28.6%)	N/A	-
E4	6 (17.1%)	N/A	-

UC—ulcerative colitis, Ctr—control, PUCAI—Paediatric activity index, inactive or mild 0–34, moderate 35–64, severe >65, Mayo endoscopic score, 0—normal colonic mucosa, 1—erythema, decreased vascular pattern, mild friability, 2—marked erythema, erosions, no vascular pattern, friability, 3—ulceration, spontaneous bleeding; Paris classification, E1—proctitis, E2—distal to splenic flexure colitis; E3—extensive, distal to hepatic flexure, E4—pancolitis; NS—not significant; N/A—not applicable.

**Table 2 jcm-11-02655-t002:** Biochemical markers of inflammation and measurement of serum and faecal MMP-9 and TIMP-1 in children with UC and Ctr.

	UC	Ctr	
CRP (mg/L)	6 (0–299.7)	0.35 (0–1.7)	*p* < 0.000
ESR (mm/h)	13 (2–84)	3 (2–12)	*p* < 0.000
WBC (×10^3^/μL)	8.22 (3.64–14.25)	6.08 (1.7–8.83)	*p* < 0.000
FCal (μg/g)	1943.7 (280.8–2950)	12.6 (3.8–42.7)	*p* < 0.000
PLT (×10^3^/μL)	336 (182–812)	238.5 (186–424)	*p* < 0.000
ALB (g/dL)	4.47 (2.17–5.05)	4.68 (4.17–5.49)	*p* = 0.001
Hb (g/dL)	12.4 (8.8–15.8)	13.35 (11.3–16.6)	*p* < 0.002
MMP-9 serum (ng/mL)	508.6 (160.2–2486.8)	322.7 (125.5–937)	*p* = 0.01
MMP-9 faeces (ng/mL)	380.4 (20.4–2459.6)	13 (2.8–140.48)	*p* < 0.000
TIMP serum (ng/mL)	175.19 (94.9–638.2)	142.05 (87.5–174.7)	*p* < 0.000
TIMP faeces (ng/mL)	25.4 (0.4–400.9)	2.9 (0.3–7.8)	*p* < 0.000

CRP—C-reactive protein, ESR—Erythrocyte sedimentation rate, WBC—White blood count, FCal—faecal calprotectin, PLT—platelet count, ALB—Albumin, Hb—haemoglobin, MMP-9 metalloproteinase-9, TIMP-1—tissue inhibitor of metalloproteinase-1, UC—ulcerative colitis, Ctr—control.

**Table 3 jcm-11-02655-t003:** Diagnostic performance of MMP-9 and TIMP-1 in differentiating UC and Ctr.

Marker	AUC	SE	95% C.I. (AUC)	*P* (AUC = 0.5)	Cut-off (ng/mL)	Sensitivity %	Specificity %
Faecal MMP-9	0.925	0.035	(0.855–0.994)	<0.000	>33.5	94.3	72.2
Faecal TIMP-1	0.881	0.045	(0.793–0.97)	<0.000	>5.8	74.3	95.0
Serum MMP-9	0.712	0.078	(0.56–0.864)	<0.006	>304.1	91.4	50.0
Serum TIMP-1	0.809	0.058	(0.695–0.922)	<0.000	>154.2	74.3	80.0

AUC—area under the curve, SE—standard error, C.I.—confidence interval, MMP-9—matrix metalloproteinase, TIMP-1—tissue inhibitor of metalloproteinase, UC—ulcerative colitis, Ctr—control.

**Table 4 jcm-11-02655-t004:** Biochemical markers of inflammation and measurement of serum and faecal MMP-9 and TIMP-1 in children with right-sided and left-sided UC.

	Right-Sided UC (E3, E4)	Left-Sided UC (E1, E2)	
CRP (mg/L)	16.59 (0.43–299.07)	3.0 (0–34.99)	*p* = 0.01
ESR (mm/h)	20.0 (2–84)	10.0 (2–25)	*p* = 0.012
WBC (×10^3^/μL)	6.08 (1.7–8.83)	6.42 (3.64–11.06)	*p* = 0.002
FCal (μg/g)	2049.0 (400.9–2673.0)	1777.5 (280.8–2950)	*p* = 0.286
PLT (×10^3^/μL)	412 (271–812)	308 (182–526)	*p* < 0.000
ALB (g/dL)	3.85 (2.17–4.78)	4.59 (3.82–5.05)	*p* < 0.000
Hb (g/dL)	10.7 (8.8–14.6)	13.1 (11.7–15.8)	*p* < 0.000
MMP-9 serum (ng/mL)	783.6 (292.1–2486.6)	492.3 (160.2–1319.1)	*p* = 0.01
MMP-9 faeces (ng/mL)	460.4 (25.7–2377.5)	372.3 (20.4–2459.6)	*p* = 0.57
TIMP serum (ng/mL)	215.6 (143.8–638.2)	157.6 (94.9–278.0)	*p* = 0.006
TIMP faeces (ng/mL)	16.8 (2.9–400.9)	43.6 (0.4–334.7)	*p* = 0.9

Paris classification, E1—proctitis, E2—distal to splenic flexure colitis, E3 extensive, distal to hepatic flexure, E4—pancolitis, CRP—C-reactive protein, ESR—Erythrocyte sedimentation rate, WBC—White blood count, FCal—faecal calprotectin, PLT—platelet count, ALB—ALBUMIN, Hb—haemoglobin, MMP-9 metalloproteinase-9, TIMP-1—tissue inhibitor of metalloproteinase-1, UC—ulcerative colitis.

**Table 5 jcm-11-02655-t005:** Diagnostic performance of MMP-9 and TIMP-1 in differentiating Right-sided UC and Left-sided UC.

Marker	AUC	SE	95% C.I. (AUC)	*P* (AUC = 0.5)	Cut-Off (ng/mL)	Sensitivity %	Specificity %
Serum MMP-9	0.75	0.084	(0.585–0.915)	=0.003	>658	56.3	84.2
Serum TIMP-1	0.77	0.081	(0.612–0.928)	<0.000	>210.9	56.3	89.5

AUC—area under the curve, SE—standard error, C.I.—confidence interval, MMP-9—matrix metalloproteinase, TIMP-1—tissue inhibitor of metalloproteinase, UC—ulcerative colitis.

**Table 6 jcm-11-02655-t006:** Significant correlations (Spearman’s rank correlation analysis) of serum TIMP-1 and serum MMP-9 with the inflammatory markers, disease activity index and endoscopic score.

		CRP	ESR	WBC	HB	PLT	ALB	FCal	PUCAI	MAYO	Paris
**Serum MMP-9**	**R**	0.443	NS	0.334	NS	0.422	NS	NS	NS	0.400	0.407
	** *p* **	0.007		0.049		0.012				0.017	0.015
**Serum TIMP-1**	**R**	0.344	0.369	NS	−0.523	0.629	−0.564	NS	0.457	NS	0.372
	** *p* **	0.043	0.034		0.001	0.000	0.000		0.006		0.027

CRP—C-reactive protein, ESR—Erythrocyte sedimentation rate, WBC—White blood count, HB—haemoglobin, ALB—albumin, FCal—faecal calprotectin, PLT—platelet count, MMP-9 metalloproteinase-9, TIMP-1—tissue inhibitor of metalloproteinase-1, PUCAI—Paediatric Ulcerative Colitis Activity Index, MAYO—Mayo endoscopic score, Paris—Paris classification for Ulcerative Colitis.

## Data Availability

The datasets generated and analysed during the current study are available from the corresponding author on reasonable request.

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
