# Peer review of "Are Matrix Metalloproteinase-9 and Tissue Inhibitor of Metalloproteinase-1 Useful as Markers in Diagnostic Management of Children with Newly Diagnosed Ulcerative Colitis?"

_jcm, 2022, doi:10.3390/jcm11092655_

Round 1

Reviewer 1 Report

Czajkowska et al. performed an interesting study about MMP-9 and TIMP utility in pediatric UC. For now, I have the following comments- 

Introduction is appropriately structured.

I wonder if the authors calculated sample size. I am generally very fond of the study protcol, as it took into consideration all the relevant aspects (serum, feces, tissue) yet I am also concerned whether appropriate conclusions can be drawn with such a small sample.

Line 291. Tendency toward significance should not be used as a term as it implies difference although there is lack of it. 

Line 292. Could the authors justify these statements

Overall discussion is written very plain and should be revised. In my opinion the authors should expand the pathophysiological background that would justify the present results. In addition, calprotectin being the most important biomarker in this setting, the authors should expand the comparison between the three, and possibly discuss that certain other biomarkers are developing in this section, such as BAFF.

Finally, as MMP-9 and TIMP were so far relatively well studied, I wonder what are the novelties of this particular study. The authors should addess it more appropriate in the section regarding strengths of the study.

Minor points

Line 50. something is wrong with this sentence

Abbreviate all p-values to 3 significant points

Figure 1 and 2 are unreadable and of very poor quality, please revise and add the appropriate p values in the figure

Author Response

Thank You for the evaluation of our manuscript. We are very grateful for Your valuable comments.

Please find our answers below.

  1. I wonder if the authors calculated sample size. I am generally very fond of the study protcol, as it took into consideration all the relevant aspects (serum, feces, tissue) yet I am also concerned whether appropriate conclusions can be drawn with such a small sample.

Answer: This was an exploratory, but not a confirmatory study where achievable effect sizes and variance can be estimated from previous exploratory evidence. We are aware that our results may be subject to type II error (low test sensitivity). In addition, it was a single-center study and within 5 years the number of newly diagnosed patients was significantly lower than the calculated sample size. During that time we were able to obtain biological material and tissue samples of most patients diagnosed with IBD, however some of them had to be excluded from the study due to therapies administered before diagnosis, blood transfusions or comorbidities. We are aware, that our results require confirmation  in further studies preceded by the calculation of sample size and test power.

  1. Line 291. Tendency toward significance should not be used as a term as it implies difference although there is lack of it. 

Answer: The sentence was corrected. “Interestingly we observed that both faecal MMP-9 and TIMP-1 were significantly higher in patients with mild lesions in the colon, assessed as 1 according to Mayo endoscopic subscore, while serum MMP-9 and TIMP-1 concentrations were higher (although only MMP-9 significantly) in a group of patients with more intensive lesions. ”

  1. Line 292. Could the authors justify these statements

Answer: Recent studies by Al-sadi et-al. proved that MMP-9 disrupts intestinal epithelial tight-junction barrier. In our study levels of serum MMP-9 were higher in the group of patients with more severe Mayo endoscopic score,. More intense inflammation in the colon is associated with a higher expression of MMP-9 in the tissue, therefore it suggest MMP-9 greater influence on intestinal barrier disruption, which leads to more intensive absorption of MMP-9 into the circulation.. The sentence was added at the end of paragraph: “This result may suggest that in the early stages of mucosal injury, when the lesions are superficial, the permeability of the colon mucosa is still effective and prevents absorption of the released enzyme. Interestingly, Al-Sadi et al. have reported an effect of MMP-9 on increased intestinal permeability, supporting our theory.”

  • Al-Sadi, R.; Engers, J.; Haque, M.; King, S.; Al-Omari, D.; Ma, T.Y. Matrix Metalloproteinase-9 (MMP-9) induced disruption of intestinal epithelial tight junction barrier is mediated by NF-κB activation. PLoS One 2021, 16, e0249544, doi:10.1371/journal.pone.0249544.
  • Al-Sadi, R.; Youssef, M.; Rawat, M.; Guo, S.; Dokladny, K.; Haque, M.; Watterson, M.D.; Ma, T.Y. MMP-9-induced increase in intestinal epithelial tight permeability is mediated by p38 kinase signaling pathway activation of MLCK gene. Am J Physiol Gastrointest Liver Physiol 2019, 316, G278-G290, doi:10.1152/ajpgi.00126.2018.

  1. Overall discussion is written very plain and should be revised. In my opinion the authors should expand the pathophysiological background that would justify the present results. In addition, calprotectin being the most important biomarker in this setting, the authors should expand the comparison between the three, and possibly discuss that certain other biomarkers are developing in this section, such as BAFF.

Answer: The discussion has been revised and the paragraphs about pathophysiology and other biomarkers (including calprotectin and BAFF) have been added (line 260-269, line 330-344).

  1. Finally, as MMP-9 and TIMP were so far relatively well studied, I wonder what are the novelties of this particular study. The authors should address it more appropriate in the section regarding strengths of the study.

Answer: In line …  “To the best of our knowledge, there is only one study measuring TIMP-1 faecal concentration in paediatric population.  (…) As far as we are aware, this is the first simultaneous measurement of TIMP-1 in serum and stool in such group of patients.” Although MMP-9 has been studied in children with IBD, research was more focused on patients with Crohn’s Disease. In our study we focused on patients with UC. The sentence “Finally, this was the only study to assess TIMP-1 levels in both serum and faeces of children with IBD, and one of two studies to assess faecal concentration of TIMP-1 in pediatric population” was added in the section regarding strengths of the study.

  1. Line 50. something is wrong with this sentence

Answer: The sentence has been changed to “Matrix Metaloproteinase-9 (MMP-9) and Tissue Inhibitor of Metaloproteinase-1 (TIMP-1) are enzymes involved in tissue remodeling and their overexpression in colonic mucosa of patients suffering from UC have been previously reported [5-8]”

  1. Abbreviate all p-values to 3 significant points

Answer: The p-values have all been abbreviated to 3 significant points.

  1. Figure 1 and 2 are unreadable and of very poor quality, please revise and add the appropriate p values in the figure

 Answer: The mentioned Figures have been replaced. P-values have been added to the Figures.

Reviewer 2 Report

It has been previously reported that the levels of MMP-9 and TIMP-1 in urine, blood and feces of UC patients were increased. Although these enzymes have been well studied in adults, there are not enough studies in pediatric patients with UC, so it is necessary to study the indicator markers in children. In this manuscript, Aleksandra et al. describe a study to determine the relation of MMP-9 and TIMP-1 with UC phenotype, disease activity index and routinely tested inflammatory markers in newly diagnosed paediatric patients. Aleksandra et al. found that the diagnostic potential of both TIMP-1 and MMP-9 measurements suggesting their clinical utility in children with UC. And they believe that sim-ultaneous measurement of these proteins both in serum and stool in combination with other common markers of inflammation could be applied in clinical practice.

The overall subject is meaningful and worthy of study. Appropriate methods were used to perform these studies. The results are extensively documented, presented in the form of figures and interpreted. Statistical analyses were also performed. Generally, the results obtained it is good. I feel that it is suitable for publication in this journal but, after the authors should accept few revisions of their paper, particularly on the following points:

L139-145, L182-192…:  It should be better to unify the letter "P" in the text, all the Tables and Figures in the full manuscript, change "P" to italic "P". e.g., P = 0.001.

L139: The unit " ng / ml " should be added under "cut off" in the Table 5.

L139-145: The groups with significant differences should be connected by connecting lines in the Figure 2, and their P value should be marked to indicate their significant differences.

L314-316: It will be better to replace the word “the assessment of studied markers in patients with other inflammatory pathologies of GI tract in children could significantly improve the relevance of the results” with “there is a lack of the assessment of studied markers in patients with other inflammatory pathologies of GI tract in children, which could significantly improve the relevance of the results”.

L321-324: It's best to delete this paragraph because it looks like a reviewer's comment.

Author Response

Thank You for the evaluation of our manuscript. We are very grateful for Your valuable comments.

Please find our answers below.

  1. L139-145, L182-192…:  It should be better to unify the letter "P" in the text, all the Tables and Figures in the full manuscript, change "P" to italic "P". e.g., P= 0.001.

Answer: The letter “P” was unifed in the text, Tables and Figures.

  1. The unit " ng / ml " should be added under "cut off" in the Table 5.

Answer: The unit was added in the Table 5.

  1. The groups with significant differences should be connected by connecting lines in the Figure 2, and their P value should be marked to indicate their significant differences.

Answer: The lines have been added and the p valuesmarked in the Figure 2.

  1. L314-316: It will be better to replace the word “the assessment of studied markers in patients with other inflammatory pathologies of GI tract in children could significantly improve the relevance of the results” with “there is a lack of the assessment of studied markers in patients with other inflammatory pathologies of GI tract in children, which could significantly improve the relevance of the results”.

Answer: The sentence was replaced: “There is a lack of the assessment of studied markers in children with other intestinal diseases, which would significantly improve the relevance of the results”.

  1. L321-324: It's best to delete this paragraph because it looks like a reviewer's comment.

Answer: The mentioned paragraph was removed.

Round 2

Reviewer 1 Report

Answer: This was an exploratory, but not a confirmatory study where achievable effect sizes and variance can be estimated from previous exploratory evidence. We are aware that our results may be subject to type II error (low test sensitivity). In addition, it was a single-center study and within 5 years the number of newly diagnosed patients was significantly lower than the calculated sample size. During that time we were able to obtain biological material and tissue samples of most patients diagnosed with IBD, however some of them had to be excluded from the study due to therapies administered before diagnosis, blood transfusions or comorbidities. We are aware, that our results require confirmation  in further studies preceded by the calculation of sample size and test power.

The authors should address this issue widely in limitation section

Answer: The sentence was corrected. “Interestingly we observed that both faecal MMP-9 and TIMP-1 were significantly higher in patients with mild lesions in the colon, assessed as 1 according to Mayo endoscopic subscore, while serum MMP-9 and TIMP-1 concentrations were higher (although only MMP-9 significantly) in a group of patients with more intensive lesions. ”

"...whereas TIMP-1 concentrations were not."

Author Response

Thank You for the evaluation of our manuscript. We are very grateful for Your valuable comments.

Please find our answers below.

  1. „Answer: This was an exploratory, but not a confirmatory study where achievable effect sizes and variance can be estimated from previous exploratory evidence. We are aware that our results may be subject to type II error (low test sensitivity). In addition, it was a single-center study and within 5 years the number of newly diagnosed patients was significantly lower than the calculated sample size. During that time we were able to obtain biological material and tissue samples of most patients diagnosed with IBD, however some of them had to be excluded from the study due to therapies administered before diagnosis, blood transfusions or comorbidities. We are aware, that our results require confirmation  in further studies preceded by the calculation of sample size and test power.” The authors should address this issue widely in limitation section

Answer: The paragraph about limitations have been changed to:

 “The study has some limitations, the most important of which is the small number of patients enrolled in it, what could have affected obtained results, especially the correlation between the studied proteins and other markers. The calculation of sample size was performed, however it was a single-center study and within 5 years the number of newly diagnosed patients was lower than the calculated sample size. Our study was an exploratory study, but not a confirmatory study where achievable effect sizes and variance can be estimated from previous exploratory evidence. We are aware that our results may be subject to type II error (low test sensitivity). That is why we did not interpret non-significant statistical results as underlying true lack of differences. We are aware, that our results need to be confirmed in further studies preceded by the calculation of sample size and test power. There is a lack of the assessment of studied markers in children with other intestinal diseases, which would significantly improve the relevance of the results.”

  1. “Answer: The sentence was corrected. “Interestingly we observed that both faecal MMP-9 and TIMP-1 were significantly higher in patients with mild lesions in the colon, assessed as 1 according to Mayo endoscopic subscore, while serum MMP-9 and TIMP-1 concentrations were higher (although only MMP-9 significantly) in a group of patients with more intensive lesions. ”"...whereas TIMP-1 concentrations were not."

Answer: The mentioned sentence has been changed to:

“Interestingly we observed that both faecal MMP-9 and TIMP-1 were significantly higher in patients with mild lesions in the colon, assessed as 1 according to Mayo endoscopic subscore, while serum MMP-9 concentrations were significantly higher in a group of patients with more intensive lesions, whereas serum TIMP-1 concentrations were not.”